# Malaria parasite CelTOS targets the inner leaflet of cell membranes for pore-dependent disruption

John R Jimah[1], Nichole D Salinas[1], Monica Sala-Rabanal[2,3], Nathaniel G Jones[1], L David Sibley[1], Colin G Nichols[2,3], Paul H Schlesinger[2], Niraj H Tolia[1,4]*

[1]Department of Molecular Microbiology, Washington University School of Medicine, Saint Louis, United States; [2]Department of Cell Biology and Physiology, Washington University School of Medicine, Saint Louis, United States; [3]Center for the Investigation of Membrane Excitability Diseases, Washington University School of Medicine, Saint Louis, United States; [4]Department of Biochemistry and Molecular Biophysics, Washington University School of Medicine, Saint Louis, United States

**Abstract** Apicomplexan parasites contain a conserved protein CelTOS that, in malaria parasites, is essential for traversal of cells within the mammalian host and arthropod vector. However, the molecular role of CelTOS is unknown because it lacks sequence similarity to proteins of known function. Here, we determined the crystal structure of CelTOS and discovered CelTOS resembles proteins that bind to and disrupt membranes. In contrast to known membrane disruptors, CelTOS has a distinct architecture, specifically binds phosphatidic acid commonly present within the inner leaflet of plasma membranes, and potently disrupts liposomes composed of phosphatidic acid by forming pores. Microinjection of CelTOS into cells resulted in observable membrane damage. Therefore, CelTOS is unique as it achieves nearly universal inner leaflet cellular activity to enable the exit of parasites from cells during traversal. By providing novel molecular insight into cell traversal by apicomplexan parasites, our work facilitates the design of therapeutics against global pathogens.

*For correspondence: tolia@wustl.edu

**Competing interests:** The authors declare that no competing interests exist.

## Introduction

Apicomplexan parasites of the genera *Plasmodium, Babesia, Cytauxzoon* and *Theileria* are responsible for the established and emerging global infectious diseases malaria (*Black et al., 2010*; *Price et al., 2007*), babesiosis (*Homer et al., 2000*), cytauxzoonosis (*Sherrill et al., 2015*), and theileriosis (*Shaw, 2003*), respectively. Although these parasites infect a range of hosts and cause different diseases, they share biological adaptations for traversal through a variety of cells within their mammalian hosts and arthropod vectors to complete their life cycle (*Homer et al., 2000*; *Sherrill et al., 2015*; *Shaw, 2003*; *Mota et al., 2001*; *Yuda and Ishino, 2004*; *Amino et al., 2008*; *Sinnis and Zavala, 2012*; *Ménard et al., 2013*; *Gueirard et al., 2010*). Traversal refers to the ability of a eukaryotic parasite to move through host or vector cells to complete essential segments of a complicated life cycle. Within the host, cell traversal by parasites is essential for successful infection as sporozoites must negotiate and cross several distinct cell types until they enter defined cells that are receptive for replication, leading to disease progression. Traversal is also essential for transmission by vectors as parasites must cross the arthropod midgut epithelia to mature into sporozoites that are the transmissible form of these apicomplexan parasites.

Traversal of parasites through host and vector cells relies on a number of different biological processes mediated by various parasite proteins. Gliding motility is the characteristic motion of

**eLife digest** Half of the world's population are at risk of contracting malaria: a disease caused by parasites that are spread by mosquito bites. Yet antimalarial drugs are becoming less and less effective because many of the parasites have grown to be resistant to them. Furthermore, and although some vaccines are already being trialed, there is not an effective vaccine that has been approved to control the disease. As such, many researchers hope that understanding more about the critical stages in the life cycle of the parasites will unveil new targets for antimalarial drugs and vaccines.

To complete their life cycle, malaria parasites must move through various cells in the human and mosquito. Parasites that lack a protein called CelTOS can enter these cells, but remain stuck inside. It was not known why this was the case, in part because the sequence of building blocks called amino acids that make up CelTOS is unlike that of other proteins which do have a known activity or function.

Jimah et al. sought to understand how CelTOS allows malaria parasites to pass through cells by solving its three-dimensional structure. This is because a protein's function often depends more on the protein's structure than the specific sequence of its amino acids; such that two proteins that have a similar shape are very likely to work in similar ways.

Jimah et al. solved the structure of CelTOS using a technique called X-ray crystallography and found that it resembles proteins known to bind and disrupt cell membranes. Further experiments showed that CelTOS from malaria parasites specifically binds to a fatty molecule that is predominantly found in the inner face of cell surface membranes. It was also confirmed that CelTOS forms pores that disrupt cell membranes. Together, these findings suggest that CelTOS breaches the cell membranes from the inside of infected human and mosquito cells to enable the parasites to exit.

While a vaccine based on the CelTOS protein is already being tested in clinical trials, the findings of Jimah et al. should enable this vaccine to be refined if it is not effective, or redesigned based on knowledge of its structure, function and mechanism. This new knowledge could also provide clues as to which kinds of molecules might neutralize the CelTOS protein's activity and therefore might lead to new antimalarial drugs.

parasites as they move through various tissues and invade cells (*Yuda and Ishino, 2004*; *Ménard et al., 2013*; *Harupa et al., 2014*; *Moreira et al., 2008*). Parasites also physically breach cell barriers during the invasion of cells (*Homer et al., 2000*; *Sherrill et al., 2015*; *Shaw, 2003*; *Mota et al., 2001*; *Yuda and Ishino, 2004*; *Amino et al., 2008*; *Kadota et al., 2004*; *Risco-Castillo et al., 2015*). Finally, parasites have developed strategies for immune evasion as they migrate through various cells, including phagocytic Kupffer cells (*Sinnis and Zavala, 2012*; *Zheng et al., 2014*). Ongoing efforts to identify and elucidate the function and mechanism of parasite proteins involved in diverse cell traversal processes will advance our understanding of parasite biology, and will unveil new targets for therapeutics. Of all the apicomplexan parasites, therapeutics for *Plasmodium falciparum* and *Plasmodium vivax* are desperately needed as these parasites are leading causes of human death and major burdens to socio-economic development (*Black et al., 2010*; *Price et al., 2007*).

<u>C</u>ell-<u>t</u>raversal protein for <u>o</u>okinetes and <u>s</u>porozoites (CelTOS) is unique among traversal proteins as it is essential for traversal of malaria parasites in both the mosquito vector and human host and is therefore critical for malaria transmission and disease pathogenesis (*Kariu et al., 2006*). Additionally, CelTOS has been identified as a promising malaria vaccine candidate referred to as Antigen 2 (*Doolan et al., 2003*). Immunization of mice with recombinant CelTOS results in humoral and cellular immune responses that reduced infection, demonstrating that targeting CelTOS is a viable approach for developing a malaria vaccine (*Bergmann-Leitner et al., 2010*; *Ferraro et al., 2013*; *Bergmann-Leitner et al., 2011*, *2013*). Even though CelTOS is critical during cell traversal by malaria parasites and is a leading transmission- and infection-blocking malaria vaccine candidate, its function remains unknown as it has no sequence similarity to proteins of known function.

In this study, we determined the structure of CelTOS in order to gain insight into its function. CelTOS structurally resembles viral membrane fusion proteins and a bacterial pore-forming toxin. This observation informed our hypothesis that apicomplexan CelTOS directly binds to and breaches plasma membranes. We show that CelTOS specifically binds phosphatidic acid, a lipid predominantly found in the inner leaflet of plasma membranes, and potently disrupts defined liposomes composed of phosphatidic acid. We determine that liposome disruption is achieved by the formation of CelTOS-dependent pores. Additionally, microinjection of CelTOS into *Xenopus* oocytes results in membrane damage. Together, these results demonstrate CelTOS is the only known apicomplexan protein with universal inner leaflet cellular activity as it addresses phosphatidic acid found specifically on the cytoplasmic face of plasma membranes, and disrupts these membranes to enable the exit of apicomplexan parasites through diverse vector and host cells. In addition to discovering this unique role of CelTOS in cell traversal, our work on the structure, function, and mechanism of CelTOS will enable structural vaccinology (*Dormitzer et al., 2012*) to produce a potent protective malaria vaccine, and inform the development of other therapeutics targeting CelTOS.

## Results

### CelTOS is conserved in apicomplexans

As CelTOS is important for multiple stages of the parasite life cycle, we examined if CelTOS was evolutionarily conserved among other apicomplexan pathogens, including those with recently sequenced genomes. We found CelTOS is conserved across various diverse branches of apicomplexan parasites including the hemosporidia (*Plasmodium* spp.) and piroplasms (*Theileria*, *Babesia*, *Cytauxzoon* spp.), groups that are thought to have diverged more than 100 million years ago (*DeBarry and Kissinger, 2011*) (*Figure 1A* and *Figure 1—figure supplement 1*). Hence, CelTOS represents an ancient and widespread adaptation that is common to apicomplexans that have two host life cycles alternating between asexual replication in their vertebrate hosts and a sexual cycle in their arthropod vectors. This is supported by the lack of CelTOS in the apicomplexan *Toxoplasma gondii* that does not require an arthropod vector for transmission. This evolutionary conservation and importance in disease pathogenesis and transmission prompted the in-depth molecular study of CelTOS that is applicable to diverse pathogenic apicomplexan parasites.

### Structure of *Plasmodium* CelTOS

The mechanism by which CelTOS enables the traversal of apicomplexan parasites through cells within the mammalian host and arthropod vector has been elusive because CelTOS has no sequence similarity to proteins of known function. Therefore, we determined the crystal structure of CelTOS from the human pathogen *P. vivax* (PvCelTOS) in order to obtain insight into function (*Figure 1*, *Figure 1—figure supplement 2* and *Figure 1—source data 1*). As CelTOS is highly conserved across apicomplexans (*Figure 1A* and *Figure 1—figure supplement 1*), the structural inferences derived from the PvCelTOS structure will apply to apicomplexan parasites generally. CelTOS forms an alpha helical dimer that resembles a tuning fork (*Figure 1B*). The dimer has a large buried surface area of 3003 Å². Consistent with the structure and large buried surface area, we established that CelTOS is an obligate dimer in solution by sedimentation equilibrium analytical ultracentrifugation (*Figure 1—source data 2*). Two N-terminal helices ($\alpha$-helices 1 and 2) of one monomer pack against the C-terminal helices ($\alpha$-helices 3 and 4) of a second monomer to form the dimer. The outer surface of the CelTOS dimer is mildly hydrophilic (*Figure 1C and E*) and masks inner hydrophobic surfaces created between the two tines of the CelTOS dimer and the dimer interface (*Figure 1E*).

### CelTOS resembles proteins that disrupt membranes

We compared the structure of the CelTOS monomer (*Figure 1D*) against all structures in the Protein Data Bank to identify proteins with similar structure and to inform function using the DALI server (*Holm et al., 2010*). Strikingly, CelTOS resembles class I viral membrane fusion glycoproteins and a bacterial pore-forming toxin with roles in membrane binding and disruption (*Figure 2*). The C-terminal helices of CelTOS show structural similarity to HIV-1 gp41 (*Figure 2A*) and *Mycobacterium bovis* ESAT-6 (*Figure 2D*), and the N-terminal helices show similarity to Hendravirus fusion core (*Figure 2B*); Nipahvirus fusion subunit core (*Figure 2C*) (*Buzon et al., 2010*; *Lou et al., 2006*;

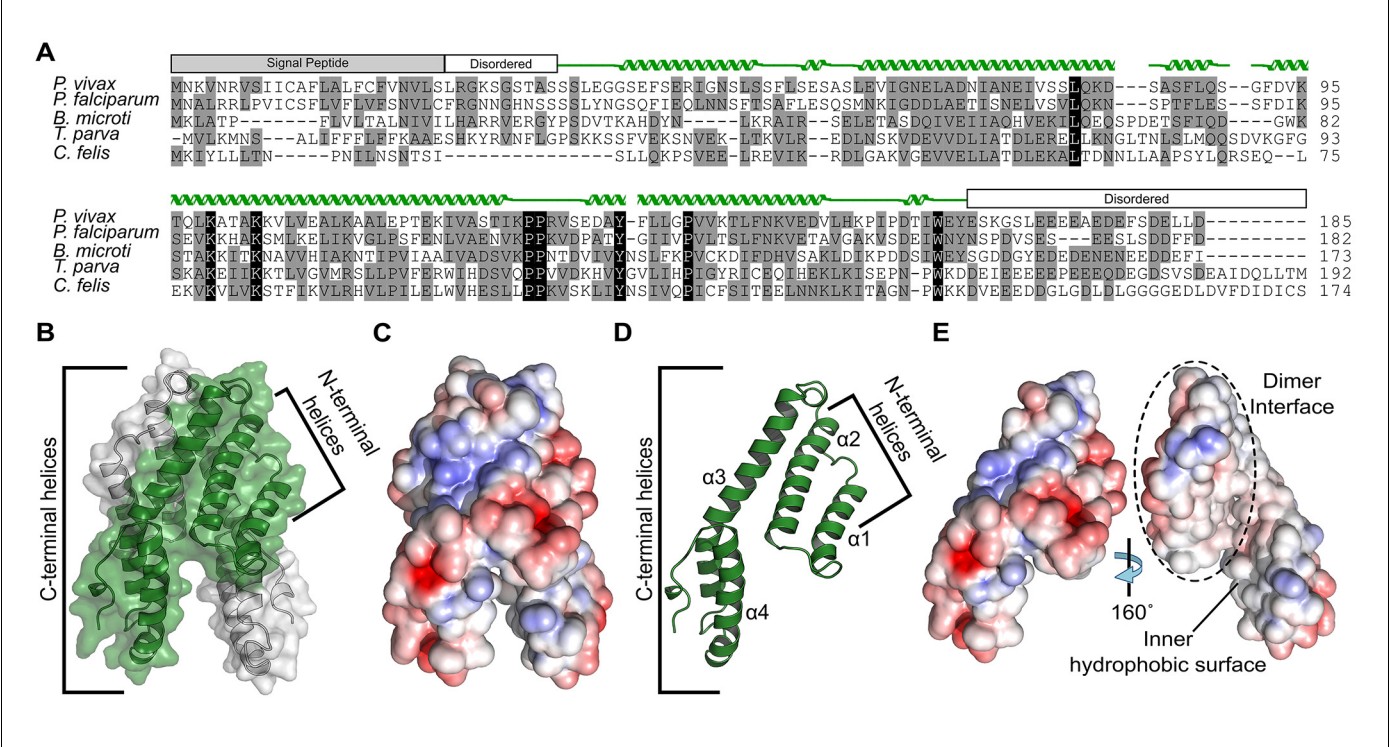

**Figure 1.** Alignment and structure of the conserved apicomplexan protein CelTOS from the human pathogen *Plasmodium vivax* (PvCelTOS). (A) Alignment of CelTOS from apicomplexan parasites *Plasmodium, Babesia, Theileria, Cytauxzoon,* and mapping of the structural elements. In the alignment, grey shading represents similarity, black shading represents identity. The secondary structure features are based on the crystal structure of PvCelTOS and shown in green. Protein accession codes as follows: *Plasmodium vivax* [UniProtKB - A5JZX5], *Plasmodium falciparum* [UniProtKB - Q8I5P1], *Babesia microti* [UniProtKB - I7J9D8], *Theileria parva* [UniProtKB - Q4N982], *Cytauxzoon felis* [PiroplasmaDB - CF003135]. (B) PvCelTOS is an alpha helical dimer that resembles a tuning fork. Each monomer is in green or white ribbon and surface representation. (C) Surface maps of CelTOS dimer showing the electrostatic surface potential colored from red ($-5$ kT e$^{-1}$) to blue (5 kT e$^{-1}$). (D) The CelTOS monomer shown as ribbon representation can be separated into two distinct subdomains composed of N-terminal ($\alpha$-helices 1 and 2) and C-terminal helices ($\alpha$-helices 3 and 4). (E) Surface maps of CelTOS monomer reveals the inner hydrophobic surfaces composed of the dimer interface and one face of the tuning fork prongs. Coloring as in *Figure 1C*.

The following source data and figure supplements are available for figure 1:

**Source data 1.** Data collection, phasing and refinement statistics.
**Source data 2.** Sedimentation equilibrium analytical ultracentrifugation analysis for Pf and PvCelTOS.
**Figure supplement 1.** Alignment of CelTOS from *Apicomplexan* parasites and mapping of the structural elements.
**Figure supplement 2.** Electron-density maps.

*Renshaw et al., 2005*; *Hsu et al., 2003*). This informed the hypothesis that *Plasmodium* CelTOS may function to bind and disrupt cell membranes, consistent with a role in host and vector cell traversal by malaria parasites. While CelTOS resembles these viral and bacterial proteins, the CelTOS structure is unique and distinct as it contains two independent subdomains that both could act as membrane disruption modules.

## CelTOS specifically binds phosphatidic acid

Given this structural similarity, we examined if CelTOS could directly bind cell membrane phospholipids and if binding was specific to a particular lipid subset using spotted arrays. Both *P. falciparum* (Pf) and *P. vivax* (Pv) CelTOS demonstrated significant binding to phosphatidic acid (PA) (*Figure 3*).

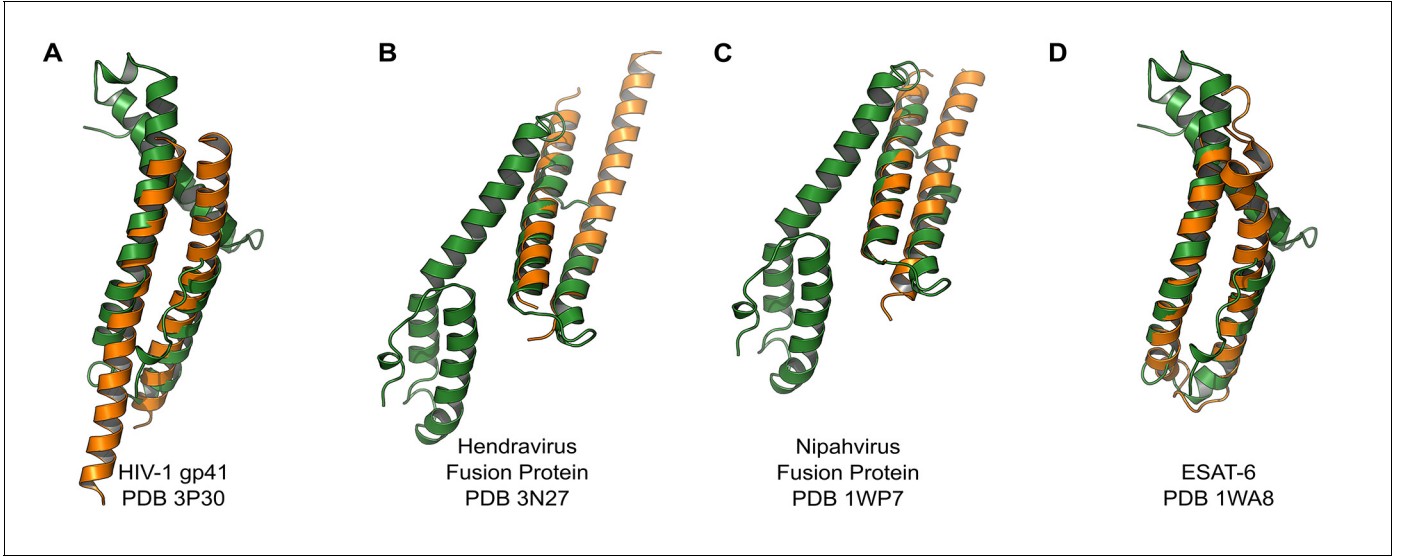

**Figure 2.** PvCelTOS is structurally similar to viral and bacterial proteins that disrupt membranes. All panels are structural overlays of PvCelTOS (green) with pathogenic membrane disrupting proteins (orange): (**A**) HIV-1 gp41 PDB 3P30 (Dali Z-score: 3.2, rmsd 3.2), (**B**) Hendravirus fusion protein PDB 3N27 (Dali Z-score: 3.1, rmsd 3.3) (orientation of CelTOS is rotated 60° along the y-axis compared to *Figure 2A*), (**C**) Nipahvirus fusion protein PDB 1WP7 (Dali Z-score: 3.3, rmsd 2.9) (orientation of CelTOS is rotated 60° along the y-axis compared to *Figure 2A*), and (**D**) *Mycobacterium bovis* ESAT-6 PDB 1WA8 (Dali Z-score: 2.6, rmsd 2.3).

The specificity to this small anionic headgroup phospholipid is unique and excludes other anionic but large headgroup phospholipids. Phosphatidic acid is predominantly found on the inner leaflet of plasma membranes (*Op den Kamp, 1979*). Specific binding of PvCelTOS to cardiolipin was also observed. However, the relevance of this binding specificity is unclear as it is not conserved in

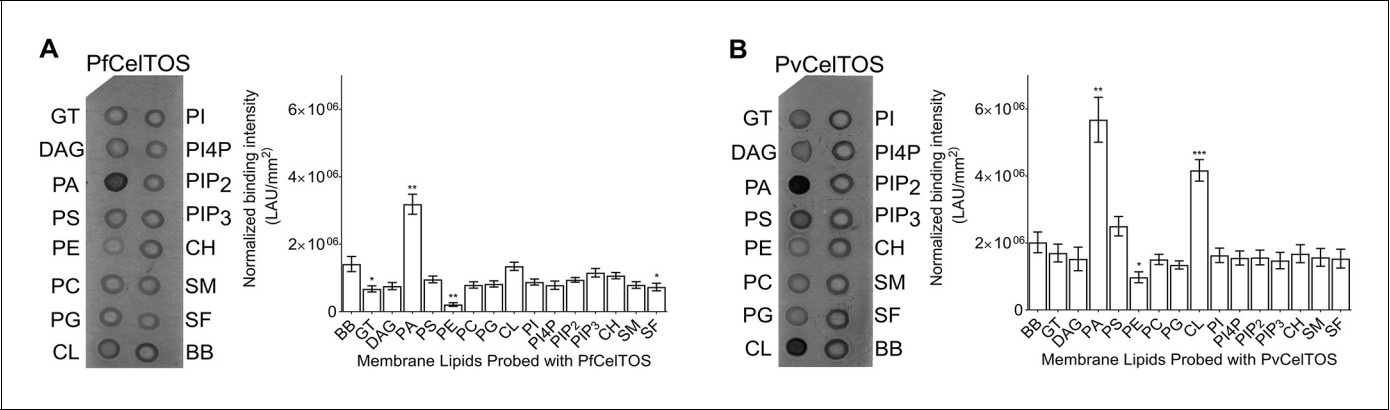

**Figure 3.** CelTOS binds the inner leaflet cell membrane lipid phosphatidic acid. (**A**) Spotted lipid array probed with PfCelTOS demonstrates significant binding to PA compared to the negative control (**BB**) in lipid arrays. Right panel: Normalized binding intensity from eight experiments shown as mean ± s.e.m. Statistical differences were determined by one-way ANOVA with matched replicates, followed by Dunnett's multiple comparison test; *$p<0.05$, **$p<0.01$, ***$p<0.001$. GT - Glyceryl tripalmitate, DAG - Diacylglycerol, PA - Phosphatidic Acid, PS - Phosphatidylserine, PE - Phosphatidylethanolamine, PC - Phosphatidylcholine, PG - Phosphatidylglycerol, CL - Cardiolipin, PI - Phosphatidylinositol, PI4P - Phosphatidylinositol (4)-phosphate, PIP$_2$ - Phosphatidylinositol (4,5)-bisphosphate, PIP$_3$ - Phoshatidylinositol (3,4,5)-trisphosphate, CH - Cholesterol, SM - Sphingomyelin, SF - 3-sulfogalactosylceramide, and BB - Blue blank. (**B**) Spotted lipid array probed with PvCelTOS demonstrates significant binding to PA and PS compared to the negative control (**BB**) in lipid arrays. Right panel: Normalized binding intensity from three experiments shown as mean ± s.e.m. Statistical differences were determined by one-way ANOVA with matched replicates, followed by Dunnett's multiple comparison test; *$p<0.05$, **$p<0.01$, ***$p<0.001$. Acronyms as in *Figure 3A*.

CelTOS from both *P. falciparum* and *P. vivax*. Limited or undetectable binding was observed to phospholipids predominantly found on the outer leaflet of plasma membranes including sphingomyelin (SM) and phosphatidylcholine (PC) (*Op den Kamp, 1979*). These results demonstrate that CelTOS has enhanced specificity for the cytosolic face of cell membranes and suggests CelTOS possesses a specific binding pocket that preferentially accommodates PA over the headgroups of other phospholipids.

## CelTOS disrupts liposomes containing phosphatidic acid

Further to identifying its lipid-binding specificity, we established that CelTOS disrupts membranes in a liposome disruption assay (*Figure 4* and *Figure 4—figure supplement 1*) (*Saito et al., 2000*). Liposomes that mimic cell membranes were created containing carboxyfluorescein at self-quenching concentrations using 1-palmitoyl-2-oleoyl-sn-glycero-3-phosphate (POPA), 1-palmitoyl-2-oleoyl-sn-glycero-3-phospho-L-serine (POPS) and 1-palmitoyl-2-oleoyl-sn-glycero-3-phosphocholine (POPC). Liposomes containing 8:2 POPC:POPA mixtures (PA liposomes) or 8:2 POPC:POPS mixtures (PS

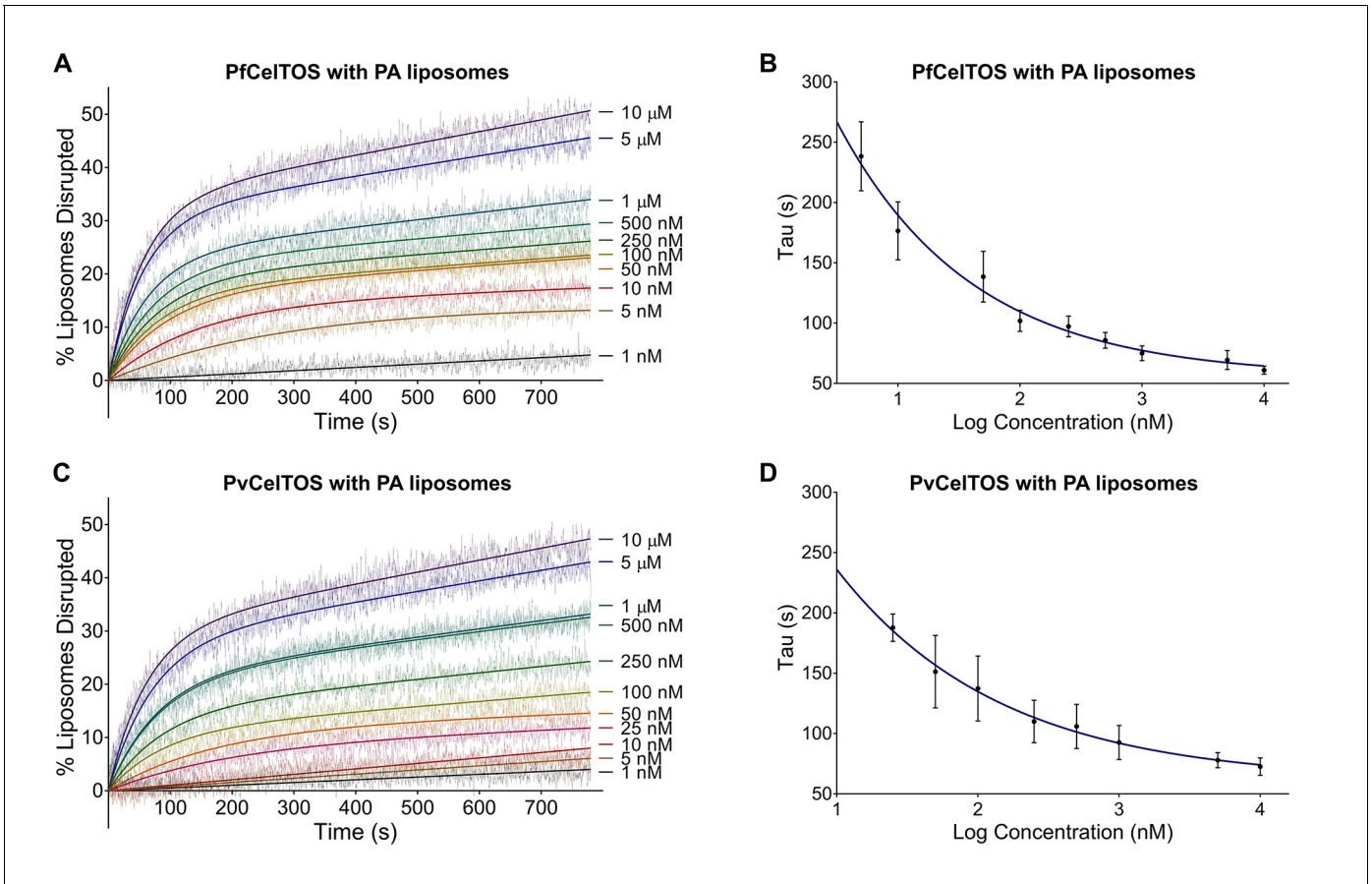

**Figure 4.** CelTOS disrupts liposomes containing phosphatidic acid (PA liposomes). (A) Time dependence of liposome disruption at various concentrations of PfCelTOS, a representative plot of three independent experiments is presented. The solid lines represent the fit of liposome disruption to *Equation 2* and the dashed lines represent the raw data of liposome disruption. (B) The time constant, Tau, determined from the fit to *Equation 2*, is plotted against the concentration of PfCelTOS for three technical replicates is shown as mean ± s.e.m. (C) Time dependence of liposome disruption at various concentrations of PvCelTOS. (D) The time constant, Tau, plotted against the concentration of PvCelTOS.

The following figure supplement is available for figure 4:

**Figure supplement 1.** CelTOS minimally disrupts liposomes composed of phosphatidylserine and phosphatidylcholine, and the negative control protein EcIsPF does not disrupt the liposomes composed of the aforementioned lipids and phosphatidic acid.

liposomes) represent the inner leaflet of plasma membranes, while pure POPC (PC liposomes) represents the outer leaflet.

Upon incubation of either Pf or PvCelTOS with PA liposomes, carboxyfluorescein was released, fluorescence dequenched, and the time-dependent increase in liposome disruption was measured (*Figure 4*).

Nanomolar concentrations of CelTOS disrupted PA liposomes consistent with a potent role in the disruption of PA containing membranes (*Figure 4*). In contrast, minimal disruption of PS and PC liposomes by CelTOS was observed and this required micromolar concentrations of CelTOS (*Figure 4—figure supplement 1A–C*). Liposome disruption was specific to CelTOS as addition of the control protein *E. coli* IspF, that has similar mass, isoelectric point and purification method as CelTOS, had no effect (*Figure 4—figure supplement 1D*). These results demonstrate that the CelTOS protein alone is necessary and sufficient for membrane disruption, specifically targets inner leaflet phosphatidic acid lipid compositions, and has a conserved function across *Plasmodium spp.*

## CelTOS disrupts liposomes by forming pores

We investigated the mechanism by which CelTOS disrupts liposomes. CelTOS could disrupt liposomes either by generalized solubilization and mixed micelle formation similar to detergents, or by forming defined protein-dependent pores within the lipid bilayer. A key test of pore formation is whether leakage of liposome contents can be prevented by plugging the pore using a molecule of the correct Stoke's radius (*Saito et al., 2000*). We examined the CelTOS-dependent release of carboxyfluorescein from liposomes in the presence of 20 μM dextran molecules of various molecular weights: 66.9 kDa (Stoke's radius:~5.8 nm), 148 kDa (Stoke's radius:~8 nm), 500 kDa (Stoke's radius: 14.7 nm), 1500–2800 kDa (Stoke's radius ranging from ~25 to ~60 nm (*Tang et al., 2016*); Stoke's radius of 2000 kDa Dextran: 27 nm [*Armstrong et al., 2004*]). Remarkably, only the 500 kDa dextran blocked carboxyfluorescein release from liposomes disrupted with PfCelTOS and PvCelTOS, respectively (*Figure 5A and B*). Dextran molecules of higher and lower molecular weights and Stoke's radius had no effect on carboxyfluorescein release, similar to experiments without dextran. Therefore, only the 500 kDa dextran has the correct Stoke's radius to occupy the pore, while the remaining dextran molecules are either too small and diffuse through the pore or too large and may not enter the pore. The 500 kDa dextran significantly lowered the maximum amount of carboxyfluorescein released disrupted (A, in *Equation 2*) compared to experiments with molecules of higher and lower molecular weights, or without dextran (*Figure 5—figure supplement 1*). This suggests that CelTOS forms pores of uniform size in liposomes and that the dextran-pore complex might be slowly reversibly tying up the protein in a non-functional pore. We visualized the CelTOS-pore in liposomes by transmission electron microscopy of negative stained liposomes (*Figure 5C*). The average diameter of the pores, reported as mean ± s.d., formed by PfCelTOS and PvCelTOS were 45.85 ± 15.55 nm and 58.62 ± 21.86 nm respectively (*Figure 5—figure supplement 2*). The dextran-blocking and electron microscopy studies suggest CelTOS forms a pore that opens a portal for parasite exit from invaded cells. The observed portal is large and strongly suggests that the membrane structure of CelTOS may continue to evolve after the initial carboxyfluorescein release.

## CelTOS disrupts cell plasma membranes by targeting the inner leaflet

The structural and activity data above suggest CelTOS functions within the cytosol of host and vector cells to disrupt cell plasma membranes from the cytoplasmic face. To test this hypothesis, we microinjected purified CelTOS into the cytosol of *Xenopus* oocytes and examined the effects on membrane integrity and cell survival (*Figure 6A and B*, and *Figure 6—figure supplement 1*). Microinjection allows for direct examination of the cellular effect of proteins. Only minor membrane phenotypes due to the injection process were observed in oocytes injected with buffer and in those injected with the negative control protein, EcIspF. In contrast, microinjection of Pf or PvCelTOS caused significant increase in the number of cells that had damaged membranes (*Figure 6A and B*, and *Figure 6—figure supplement 1*).

We examined if CelTOS can disrupt cell membranes from the extracellular face using RAW 264.7 macrophages. The cell membrane of *Xenopus* oocytes is protected from the extracellular environment by a vitelline membrane. This makes oocytes unsuitable for studying the effect of CelTOS on the outer surface of cell membranes. RAW macrophages have an exposed cell membrane and are

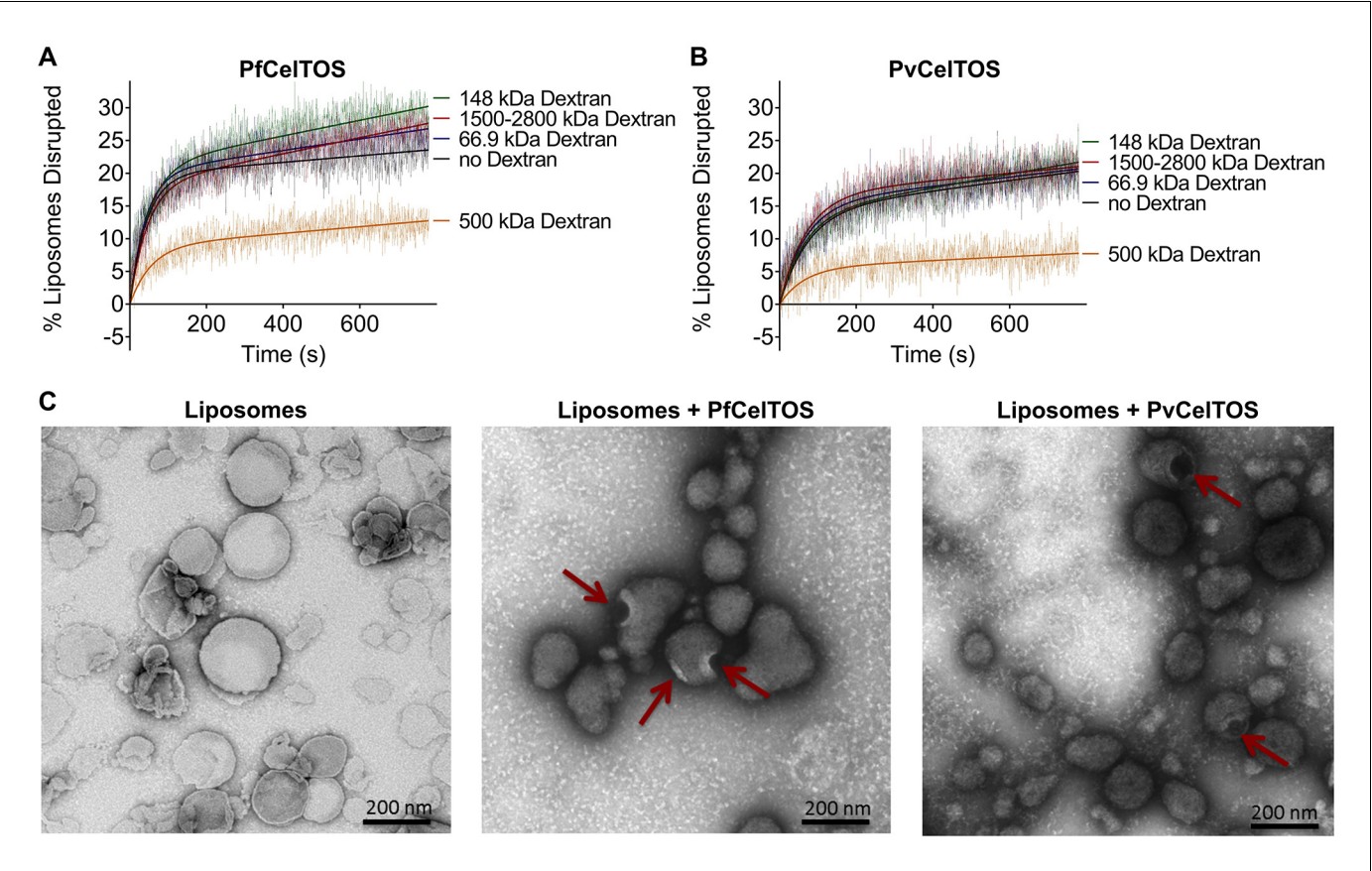

**Figure 5.** CelTOS forms pores in lipid membranes and a 500 kDa Dextran (Stoke's radius = 14.7 nm) blocks the release of carboxyfluorescein from PA liposomes disrupted by PfCelTOS and PvCelTOS. Time dependence of PA liposome disruption by (**A**) PfCelTOS or (**B**) PvCelTOS at 250 nM in the presence of dextran molecules of various molecular weights, a representative plot is presented. The solid lines represent the fit of liposome disruption to *Equation 2* and the dashed lines represent the raw data of liposome disruption. (**C**) Transmission electron microscopy of negative stained liposomes alone (left), liposomes treated with PfCelTOS (middle), and liposomes treated with PvCelTOS (right) reveals CelTOS-dependent pores (red arrows).

The following figure supplements are available for figure 5:

**Figure supplement 1.** Quantitation of the dextran block experiment.

**Figure supplement 2.** Diameters of pores in liposomes formed by CelTOS.

an especially relevant model as they are similar to liver macrophages (Kupffer cells), which are traversed by malaria parasites in a CelTOS-dependent process. No membrane disruption, measured by uptake of a membrane-impermeable fluorescent dye, was observed for macrophages in tissue culture when CelTOS was added to the extracellular media (*Figure 6C*). This is fully consistent as CelTOS does not bind phospholipids found predominantly in the outer leaflet of plasma membranes, rendering it incapable of disrupting plasma membranes from the extracellular face (*Figure 3*, *Figure 6C* and *Figure 4—figure supplement 1C*).

## A model for CelTOS during cell traversal

This study provides unprecedented insight into the structure, function and mechanism of a critical protein involved in host and vector cell traversal by apicomplexan parasites. We have identified CelTOS as the only parasite protein known to breach plasma membranes from the cytoplasmic face to enable the exit of parasites from cells during traversal. The data from this study support the model depicted in *Figure 7*. CelTOS is released from parasite secretory organelles and adheres to the surface of the malaria parasite prior to and during the traversal of host and vector cells (*Bergmann-*

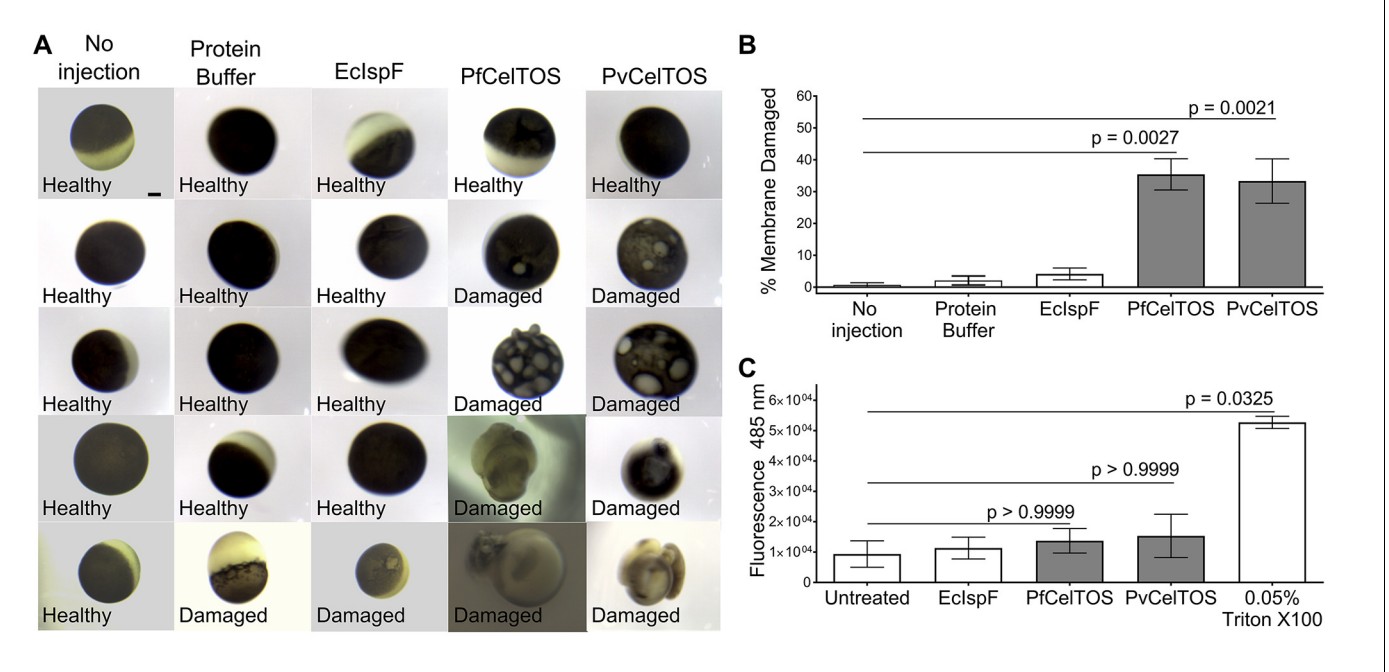

**Figure 6.** CelTOS disrupts cells by directly binding the cytosolic face of cell plasma membranes. (**A**) Microinjection of *Xenopus* oocytes with CelTOS results in membrane damage. Cells were incubated at room temperature and examined 24 hr post-injection. Five representative images from 24 individually microinjected oocytes per condition are shown. Scale bar represents 0.2 mm. (**B**) Percent of oocytes with damaged membranes from six independent experiments each with 24 technical replicates. Statistical differences were determined by Kruskal-Wallis and Dunn's multiple comparison tests. The graph is shown as mean ± s.e.m. (**C**) RAW 264.7 macrophages used in a cytotoxicity assay demonstrate CelTOS is unable to disrupt cells from the extracellular surface. Membrane damage was measured by uptake of a membrane-impermeable fluorescent dye. Statistical differences were determined for three independent experiments each with three technical replicates by Kruskal-Wallis and Dunn's multiple comparison test. The graph is shown as mean ± s.e.m.

The following figure supplement is available for figure 6:

**Figure supplement 1.** Complete images for a representative experiment of Xenopus oocyte microinjection.

*Leitner et al., 2010*, *2011*). Our results show that when the parasite is extracellular, CelTOS has limited cell membrane disrupting activity due to its low preference for lipids on the outer surface of plasma membranes. Once the parasite invades the cytosol of host and vector cells, CelTOS binds with high specificity to lipids in the inner leaflet of the plasma membrane. Binding leads to localized membrane disruption and pore formation that creates a portal to ultimately direct the exit of the parasite from the host or vector cells to complete traversal.

## Discussion

Traversal of apicomplexan parasites across cells of their arthropod vectors and vertebrate hosts is required for transmission and disease progression (*Homer et al., 2000*; *Shaw, 2003*; *Mota et al., 2001*; *Yuda and Ishino, 2004*; *Amino et al., 2008*; *Sinnis and Zavala, 2012*). Parasites utilize gliding motility during traversal and have the ability to breach plasma membranes from the extracellular space to invade cells. Parasite proteins involved in gliding motility (*Yuda and Ishino, 2004*; *Tardieux and Ménard, 2008*), and the breaching of plasma membranes from the extracellular space to enable entry (*Kadota et al., 2004*; *Risco-Castillo et al., 2015*; *Ishino et al., 2005*) have been reported. The current literature supports a model where perforin-like proteins (PLPs) mediate entry into various pre-erythrocytic cells including the PLP-1 dependent disruption of transient vacuoles created during entry of hepatocytes (*Risco-Castillo et al., 2015*; *Ishino et al., 2005*). This suggests a mechanism in which PLPs engage outer leaflet lipids for disruption, although direct studies probing

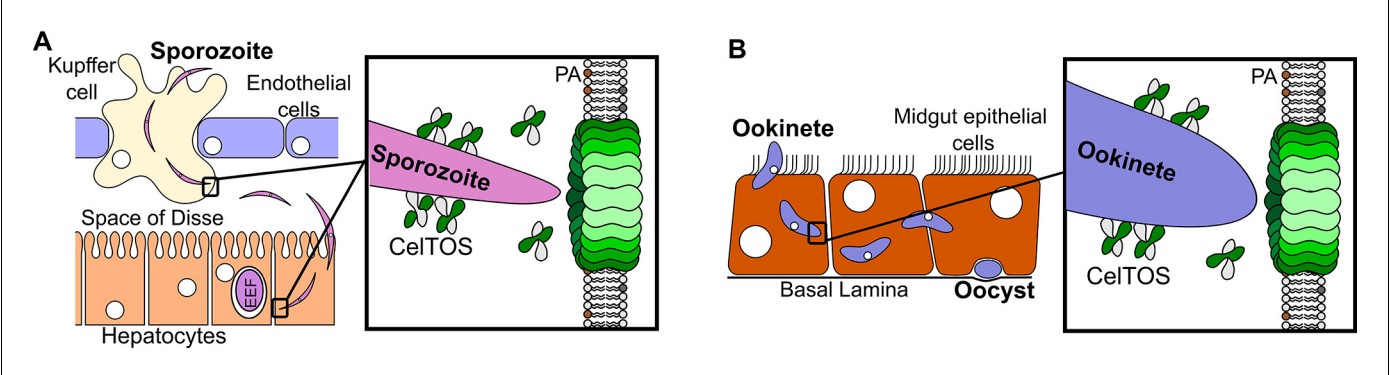

**Figure 7.** Model for CelTOS in cell traversal. (**A**) During the pre-erythrocytic stage, malaria parasite sporozoites traverse various host cells including Kupffer cells and hepatocytes. CelTOS forms pores and disrupts the cell membranes of these cells to allow exit of the sporozoites to complete traversal. The inset shows CelTOS is localized to the surface of sporozoites and disrupts plasma membranes by directly binding to phosphatidic acid (PA), colored brown, in the inner leaflet to create a pore that enables sporozoite exit. EEF – exo-erythrocytic form. (**B**) During the mosquito stage, malaria parasite ookinetes traverse the mosquito midgut epithelium to reach the basal lamina where it develops into oocysts. CelTOS forms pores and disrupts vector cell membranes to direct ookinete exit during traversal through the mosquito midgut epithelium. The inset shows CelTOS is localized to the surface of ookinetes and disrupts plasma membranes by directly binding to phosphatidic acid (PA), colored brown, in the inner leaflet to create a pore that enables ookinete exit.

the biochemical function and lipid specificity of PLPs have not been described. Further, PLPs have been shown to play a role in egress of parasites from the parasitophorous vacuole and red blood cells (*Garg et al., 2013*; *Wirth et al., 2014*). However, proteins involved in breaching the plasma membrane of invaded cells during traversal, from the cytoplasmic face, to enable the exit of parasites are unknown.

We have discovered that secreted apicomplexan CelTOS attacks plasma membranes from the cytoplasmic face. CelTOS had previously been identified as a malaria antigen, and CelTOS-deficient parasites are unable to traverse cells of their host and vector (*Kariu et al., 2006*; *Doolan et al., 2003*). The results of our study indicate that CelTOS functions to breach plasma membranes from the cytoplasmic face to enable the exit of parasites from invaded host and vector cells during traversal. This study challenges the paradigm that motility alone is sufficient for parasite exit, as it is now clear that active membrane disruption by CelTOS is necessary for parasite release. This work identifies CelTOS-dependent pore formation leading to inner-leaflet membrane disruption as a new active molecular process during traversal, and presents novel avenues for preventing traversal by inhibiting inner leaflet membrane disruption. As CelTOS is conserved across several large groups of apicomplexan parasites including *Plasmodium spp.*, *Cytauxzoon*, *Theileria* and *Babesia*, this finding informs fundamental pathogen biology applicable to multiple diseases and microorganisms of global importance with particular relevance to cell infection, traversal and membrane disruption.

While CelTOS resembles viral membrane fusion proteins and a bacterial pore-forming toxin, our findings define a new class of membrane disruption proteins. CelTOS has a distinct structural architecture with two subdomains that independently resemble membrane binding and/or disrupting proteins and could simultaneously act during disruption (*Figure 2*). Further, CelTOS employs a unique lipid specificity for phosphatidic acid to achieve nearly universal inner leaflet cellular activity. This new class is also supported by the fact that primary sequence similarity searches did not identify membrane disruption proteins as relatives of CelTOS.

Both the N-terminal and C-terminal helices in CelTOS resemble the heptad repeats that form helices in class I viral fusion proteins (*Figure 2A–C*) (*Sackett and Shai, 2002*; *Harrison, 2015*; *LeDuc and Shin, 2000*). Viral heptad repeats undergo conformational changes that enhance the destabilization of host membranes post-insertion of the viral fusion peptide that enables lipid mixing during fusion(38). It is plausible that the N- and C-terminal helices in CelTOS may also undergo similar conformational changes during membrane disruption. However, CelTOS employs a distinct membrane disruption mechanism from viral proteins as lipid mixing is not required for CelTOS to target inner leaflet lipids and disrupt plasma membranes.

In addition, CelTOS is structurally similar to the pore-forming toxin *M. tuberculosis* ESAT-6 (*Figure 2D*) (*Hsu et al., 2003*; *Los et al., 2013*; *Smith et al., 2008*). However, these toxins generally recognize a host cell target or receptor embedded within the target membrane, and ESAT-6 is secreted in complex with a chaperone that maintains it in an inactive state. Upon binding the membrane target, the toxin sheds the chaperone, oligomerizes and embeds into the membrane to form a pore on the angstrom length scale for the passage of ions and water (*Los et al., 2013*). In contrast to a membrane-embedded target or receptor, CelTOS specifically binds phosphatidic acid within the inner leaflet of host or vector cell membranes. The dimeric nature of CelTOS may serve as a chaperone-like function to retain CelTOS in an inactive state prior to membrane binding as the dimer masks the inner hydrophobic faces of CelTOS. We provide evidence from dextran block experiments and transmission electron microscopy that CelTOS forms a pore of uniform size in liposomes.

Malaria continues to be a global health problem, with half of the world's population at risk. The lack of an effective vaccine against malaria, and the continuous evolution of drug-resistant parasites make malaria control difficult. Targeting the liver-stage (*Good and Doolan, 1999*) and mosquito-stages (*mCGo, 2011*) are of high priority for malaria vaccine development. Immunization of human volunteers with radiation-attenuated sporozoites induces sterile immunity (*Good and Doolan, 1999*; *Hoffman et al., 2002*), and all volunteers showed strong immune responses to CelTOS (referred to as Antigen 2) that were correlated with protection (*Doolan et al., 2003*). In contrast, circumsporozoite protein, which is the major component of the most advanced malaria vaccine RTS,S (*Crompton et al., 2010*), was only recognized by a third of the volunteers and mostly in those who were not protected (*Doolan et al., 2003*). Individuals living in malaria endemic regions mount IFN-γ responses to CelTOS suggesting that CelTOS is a target of protective immunity (*Anum et al., 2015*). CelTOS is also unique as it is a promising multi-stage malaria vaccine target (*Bergmann-Leitner et al., 2010*; *Ferraro et al., 2013*; *Bergmann-Leitner et al., 2011*, *2013*) for both infection-blocking and transmission-blocking vaccines currently in clinical trials.

We show that CelTOS has conserved critical function in both *Plasmodium falciparum* and *Plasmodium vivax* that infect humans. The high-sequence similarity between Pf and PvCelTOS suggest that CelTOS may elicit cross-species protection. Therefore, CelTOS could be used alone or in combination with other antigens already in development to produce an effective multicomponent vaccine which targets both the mosquito and human stages and may be cross-species protective (*Doolan et al., 2003*; *Crompton et al., 2010*; *Crosnier et al., 2011*). CelTOS-based vaccines have demonstrated the ability to elicit protection from malaria infection. However, protection is not absolute suggesting improvements in the immunogenicity, protectivity and formulation of CelTOS-based vaccines are necessary. Structure-function studies not only provide mechanistic insight into host-pathogen interactions (*Tolia et al., 2005*; *Batchelor et al., 2011*; *Vulliez-Le Normand et al., 2012*; *Doud et al., 2012*; *Malpede et al., 2013*; *Batchelor et al., 2014*; *Wright et al., 2014*; *Lin et al., 2012*), but also provide the necessary framework for improved vaccine design through structural vaccinology (*Dormitzer et al., 2012*; *Chen et al., 2013*, *2015*, *2016*). Immune recognition depends on antigen conformation and on surface accessibility of individual residues. The structure and architecture of CelTOS provides necessary insight to inform immune recognition, and enables protein engineering to retain the structural fold during immunogen design. The functional and mechanistic analyses presented here will enable assaying of immune responses for their ability to disrupt CelTOS function and provide an accurate molecular correlate for protection. Collectively, this work enables rational approaches to block CelTOS function, and rational approaches for vaccine development of global and emerging infectious diseases.

## Materials and methods

### Alignment of CelTOS from diverse apicomplexan parasites

CelTOS homologs in diverse apicomplexan parasites were identified using jackhammer and examination of the EuPathDB, and aligned in ClustalW.

## Protein expression and purification

Codon-optimized PfCelTOS amino acids 25–182, PvCelTOS amino acids 36–196 and EcIspF amino acids 1–159 with C-terminal hexahistidine tags, were expressed in *E. coli*. Tagged proteins were purified by Nickel-NTA chromatography and gel filtration using a Superdex 200 10/300 GL column (GE Healthcare Life Sciences, Pittsburg, PA).

## Crystallization, data collection and structural studies

PvCelTOS crystals were grown at 17°C by hanging-drop vapor diffusion after mixing 1 µl of protein at 20 mg/ml with 1 µl of reservoir containing 1.6 M Ammonium dihydrogen phosphate, 0.08 M Tris pH 8.5 and 20% glycerol and streak seeding PvCelTOS crystals previously obtained from the same crystallization condition. Native PvCelTOS crystals were harvested and soaked for ~1 min in cryo-protectant solution containing: 1.6 M Ammonium dihydrogen phosphate, 0.08 M Tris pH 8.5 and 30% glycerol. Crystals of the PvCelTOS Platinum- derivative were obtained by soaking native Cel-TOS crystals for 1–2 min in cryo-protectant solutions containing: 10 µM Potassium tetracyanoplatinate$_{(II)}$ hydrate, 1.6 M Ammonium dihydrogen phosphate, 0.08 M Tris pH 8.5 and 30% glycerol. Crystals of the CelTOS Mercury- derivative were obtained by soaking native CelTOS crystals for 1–2 min in cryo-protectant solutions containing: 1 µM mercury ethylphosphate, 1.6 M Ammonium dihydrogen phosphate, 0.08 M Tris pH 8.5 and 30% glycerol. The structure was phased in SHARP/autoSHARP (*Vonrhein et al., 2007*) as multiple isomorphous replacement with anomalous scattering. Phases were extended by solvent flattening in SHARP/autoSHARP. The structure was refined in PHENIX (*Adams et al., 2002*) with NCS restraints. Map and model manipulation was aided by the CCP4 program suite (*Collaborative Computational Project No. 4, 1994*). MolProbity (*Davis et al., 2007*) places this structure in the top 100th percentile of structures with comparable resolution. 98.44% and 1.56% of residues lie in the favored and allowed regions of the Ramachandran plot, respectively. Structurally similar PDBs were identified using the Dali server (*Holm et al., 2010*), and alignments and the normalized rmsd values obtained using lsqman (*Kleywegt, 1996*). Buried surface area of the dimer was determined using PDBePISA (*Krissinel and Henrick, 2007*). Atomic coordinates and structure factors for the reported crystal structure were deposited into the Protein Data Bank under PDB code 5TSZ (http://www.pdb.org).

## Analytical ultracentrifugation

Sedimentation equilibrium experiments were conducted with a Beckman/Coulter XL-A analytical ultracentrifuge (Beckman/Coulter, Indianapolis, IN) using an An60Ti rotor at 10°C and λ 286 nm. PfCelTOS and PvCelTOS were purified by gel filtration using a Superdex 200 10/300 GL column (GE Healthcare Life Sciences) in 150 mM NaCl and 10 mM HEPES pH 7.4. Data were collected for speeds 12,000 rpm and 15,000 rpm with PfCelTOS at 25 µM, 36 µM and 46 µM, and PvCelTOS at 29 µM, 41 µM and 53 µM. A partial specific volume of 0.730919 and 0.737102 was calculated using Sednterp for PfCelTOS and PvCelTOS, respectively. Data were analyzed using a single component model with UltraScan II version 9.9. Average molecular weight is reported as mean ± s.d.

## Membrane lipid screen

Membrane lipid strips (Echelon Biosciences Inc., Salt Lake City, UT) were probed with proteins according to the manufacturer specified protocol. Membrane strips were blocked at room temperature for 1 hr in blocking buffer (10 mM Tris pH 8.0, 150 mM NaCl, 0.1% Tween 20% and 3% BSA) and washed three times for 5 min with wash buffer (10 mM Tris pH 8.0, 150 mM NaCl and 0.1% Tween 20). PfCelTOS and PvCelTOS were incubated in blocking buffer at a concentration of 20 µM. Mouse anti-His monoclonal antibody (Invitrogen, Waltham, MA) diluted 1:500 in blocking buffer was used as the primary antibody, and Peroxidase AffiniPure Goat Anti-Mouse IgG (Jackson Laboratories) diluted 1:10,000 in blocking buffer as the secondary antibody. Binding was detected with ECL Prime Western Blotting Detection Kit (GE Healthcare Biosciences), imaged using Image Reader FLA-5000 series phosphorimager (Fuji PhotoFilm, Japan) and quantified using Image Gauge version 4.23 (Fuji PhotoFilm). Experiments were performed with eight replicates, along with one negative control where CelTOS was omitted. Spot intensities were normalized to background and the negative control strip. Data were analyzed by one-way ANOVA with Dunnett's multiple comparison tests to compare the intensity of each lipid spot to the negative control on that strip using Prism 6 (GraphPad

Software, La Jolla, CA). The normalized binding intensity of each spot was plotted as mean ± s.e.m. The membrane lipid screen is described in more detail at Bio-protocol (*Jimah et al., 2017a*).

## Liposome preparation and liposome disruption assay

Liposomes used for experiments were composed of 1-palmitoyl-2-oleoyl-sn-glycero-3-phosphocholine (POPC) alone or in combination with either 1-palmitoyl-2-oleoyl-sn-glycero-3-phosphate (POPA) or 1-palmitoyl-2-oleoyl-sn-glycero-3-phospho-L-serine (POPS) (Avanti Polar Lipids, Alabaster, AL). Chloroform-dissolved lipids were dried under $N_2$ gas followed by vacuum for 3 hr. To hydrate the lipids, equal volumes of ether and a solution composed of 10 mM HEPES pH 7.4, 150 mM KCl and 20 mM carboxyfluorescein were added to the lipids, the mixture sonicated and ether evaporated using a roto-vap. Liposomes were sized by extrusion using 200 nm polycarbonate Track-Etched filters (Whatman/Nucleopore). Size exclusion chromatography with Sephadex G 25–300 (Sigma Aldrich, Saint Louis, MO) was used to separate unincorporated carboxyfluorescein from liposomes. In all experiments, varying concentrations of proteins were incubated with liposomes containing 250 nM of lipids, and the time-dependence of liposome disruption was monitored using a Cary Eclipse Fluorescence Spectrophotometer (Varian Inc/ Agilent, Santa Clara CA) by observing the carboxyfluorescein fluorescence emission at 512 nm upon excitation at 492. Three replicate measurements were performed for each protein. The percent liposome disruption was calculated as:

$$\%\mathrm{Disruption_{time}} = [(\mathrm{F_{512\,of\,liposome+protein}} - \mathrm{F_{512\,of\,liposome}})/(\mathrm{F_{512\,of\,liposome+triton}} - \mathrm{F_{512\,of\,liposome}})]^* 100 \quad (1)$$

where 0.2% Triton X-100 was added to normalize each sample to complete dequenching as previously described, and data fitted to *Equation 1* (*Saito et al., 2000*).

For kinetic and pore-forming analyses, the time dependence of liposome disruption was also fitted to the following *Equation 2* (*Saito et al., 2000*):

$$\%\mathrm{Liposomes\,Disrupted} = \mathrm{A}(1 - \mathrm{e}^{-(\mathrm{time/tau})}) + \mathrm{m}^*\mathrm{time} \quad (2)$$

where **A** is the maximum percentage of liposomes disrupted, **Tau** is the time constant for the exponential component, and **m** is the slope of the linear component.

For the liposome disruption assays with dextran, 20 µM dextran was added to liposomes, and then 250 nM of PfCelTOS or PvCelTOS was added, and liposome disruption monitored. The maximum amount of liposome disrupted, A, was obtained by fitting the data to *Equation 2*. Data was analyzed using Prism 6 (GraphPad Software, La Jolla, CA) by performing a one-way ANOVA with Dunnett's multiple comparison test. Data with seven experimental replicates of each preparation included as mean ± s.e.m. The membrane lipid screen is described in more detail at Bio-protocol (*Jimah et al., 2017b*).

## Microinjection of *Xenopus laevis* oocytes with *Plasmodium* CelTOS

Mature female *Xenopus laevis* frogs were purchased from *Xenopus* Express (Brooksville, FL). All animal protocols followed guidelines approved by the Washington University School of Medicine and the National Institutes of Health. Frogs were anesthetized with a 0.1% tricaine solution buffered with 0.1% $NaHCO_3$ prior to survival surgery in which a portion of the ovary is removed. Stage V-VI oocytes were isolated and maintained at 18 in modified Barth's solution of the following composition: 88 mM NaCl, 2.4 $NaHCO_3$, 1 mM KCl, 0.3 mM $Ca(NO_3)_2$, 0.4 mM $CaCl_2$, 0.8 mM $MgSO_4$ and 10 mM HEPES/Tris pH 7.4, and supplemented with 50 mg/l gentamicin, 6 mg/l ciprofloxacin, and 100 mg/l streptomycin sulfate/100,000 units/l penicillin G sodium (Life Technologies, Carlsbad, CA). One day after isolation, oocytes were microinjected with 50 nl (0.1 nmol, approximately 100 µM final concentration) of EcIspF, PfCelTOS or PvCelTOS, prepared fresh as 2 mM stock solutions in Protein buffer composed of 10 mM HEPES pH 7.4 and 150 mM KCl; non-injected oocytes, oocytes injected with 50 nl of protein buffer, and oocytes injected with EcIspF, served as controls. Twenty-four oocytes per condition were individualized in 96-well plates, and monitored for membrane damage. Oocytes were monitored 24 hr post-infection under an Olympus SZX10 microscope (Olympus Corporation, Waltham, MA), and images captured with Infinity Lite camera using Infinity capture version 6.00 software (Lumnera Corporation). Results are shown as mean ± s.e.m. of six separate experiments with independent protein preparations and with oocytes from different donor frogs. Data

were analyzed using Prism 6 (GraphPad Software, La Jolla, CA) by performing Kruskal-Wallis followed by Dunn's multiple comparison tests.

### RAW 264.7 macrophage cytotoxicity assay

RAW 264.7 cells (ATCC) liberated using CellStripper reagent (Corning, Corning, NY) were filtered through a 40 µm mesh to yield a single-cell suspension. Cells were seeded onto clear-bottomed, white-walled 96-well plates (Corning 3610) at $2 \times 10^5$ cells per well in DMEM 10% FBS and allowed to adhere overnight. Wells were washed with PBS and assay reagents added. The assay was conducted in 5% FBS with phenol-red-free DMEM. Recombinant proteins were added to 100 µM final concentration. Triton X-100 at a final concentration of 0.05% v/v served as a positive control. Total reaction volume was 100 µl. The plates were incubated at 37°C, 5% $CO_2$ for 3.5 hr prior to the addition of 50 µl 1x CellTox Green reagent (Promega, Madison, WI) and incubated for a further 30 min. Fluorescence at 485 nm was read using a Cytation 3 Cell Imaging Multimode Reader (Biotek, Winooski, VT). Results are shown as the mean ± s.e.m. of three separate experiments with independent RAW 264.7 cells each with three technical replicates. Data were analyzed using Prism 6 (GraphPad Software, La Jolla, CA) by performing Kruskal-Wallis and Dunn's multiple comparison tests.

### Transmission electron microscopy

Liposomes were treated with either 250 µM PfCelTOS or PvCelTOS. Suspension of liposomes were then allowed to absorb onto freshly glow discharged formvar/carbon-coated copper grids for 10 min. Grids were washed in $dH_2O$ and stained with 1% aqueous uranyl acetate (Ted Pella Inc., Redding, CA) for 1 min. Excess liquid was gently wicked off and grids were allowed to air dry. Samples were viewed on a JEOL 1200EX transmission electron microscope (JEOL USA, Peabody, MA) equipped with an AMT megapixel digital camera (Advanced Microscopy Techniques, Woburn, MA). Pore diameters were determined using ImageJ 1.48V.

## Acknowledgements

This work was supported by the National Institutes of Health (R56 AI080792 to NHT) and the Burroughs Wellcome Fund (to NHT). We thank J Nix and ALS Beamline 4.2.2 supported by contract DE-AC02-05CH11231. We thank W Beatty and B Anthony of the Molecular Microbiology Imaging Facility, School of Medicine, Washington University in St Louis. The authors declare there are no conflicts of interest.

## Additional information

### Funding

| Funder | Grant reference number | Author |
| --- | --- | --- |
| Burroughs Wellcome Fund | | John R Jimah<br>Nichole D Salinas<br>Niraj H Tolia |
| National Institutes of Health | R56 AI080792 | Niraj H Tolia |

The funders had no role in study design, data collection and interpretation, or the decision to submit the work for publication.

### Author contributions

JRJ, MS-R, NHT, Conception and design, Acquisition of data, Analysis and interpretation of data, Drafting or revising the article; NDS, NGJ, Acquisition of data, Analysis and interpretation of data, Drafting or revising the article; LDS, CGN, Analysis and interpretation of data, Drafting or revising the article; PHS, Conception and design, Analysis and interpretation of data, Drafting or revising the article

### Author ORCIDs

Niraj H Tolia, http://orcid.org/0000-0002-2689-1337

## Ethics

Animal experimentation: All procedures followed guidelines approved by the Washington University School of Medicine and the National Institutes of Health, as outlined in Washington University Animal Studies Committee protocol number 2015-0110.

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
