## [Decision Letter]

Thank you for submitting your article "Malaria parasite CelTOS targets the inner leaflet of cell membranes for pore-dependent disruption" for consideration by *eLife*. Your article has been favorably evaluated by Richard Losick (Senior Editor) and three reviewers, one of whom, Elena Levashina, is a member of our Board of Reviewing Editors. One of the other reviewers, Olivier Silvie, has agreed to reveal his identity.

The reviewers have discussed the reviews with one another and the Reviewing Editor has drafted this decision to help you prepare a revised submission.

Summary:

In apicomplexan parasites, CelTOS is a critical protein required during cell traversal, and is a leading malaria vaccine candidate. CelTOS has no sequence similarity to proteins of known function, and its function remained unknown. In this study, the authors present an X-ray crystal structure of CelTOS from Plasmodium vivax. This protein is also found in several parasites other than Plasmodium providing a relevant model for other Apicomplexa. The structural data reveal that CelTOS forms an α helical dimer, and structurally resembles class I viral membrane fusion proteins and a bacterial pore-forming toxins. The authors further show that CelTOS specifically binds lipids predominantly found in the inner leaflet of plasma membranes (phosphatidic acid and phosphatidylserine), and potently disrupts liposomes of defined composition by the formation of CelTOS-dependent pores. Additionally, the authors show that CelTOS induces membrane damage in microinjected *Xenopus oocytes*, but has no lytic activity on the outer membrane leaflet of RAW cells. Together, the results demonstrate that CelTOS has a universal inner leaflet cellular activity, and strongly support a role of CelTOS in inner membrane disruption to enable the exit of apicomplexan parasites through diverse vector and host cells.

This study is very well performed and provides novel and important insights into the structure, function and mechanism of a critical protein involved in host and vector cell traversal by apicomplexan parasites. In order to improve the manuscript, the authors may consider addressing the following points.

Essential revisions:

1) The crystallographic work appears competently executed, yet insufficiently described for experiments to be repeated. For example, in the subsection “Crystallization, data collection, and structural studies”: Mercury and platinum derivative data, the authors should be more specific, and give more details (compound formula, concentration, soak duration).

2) The authors interpretation of their work depends on CelTOS forming an oligomer on the membrane but they provide no data to show this. The protein is a dimer in the crystal and in solution (AUC data). Does it act as a dimer or do the authors think it comes apart to act? Some other pore-forming proteins are constitutive dimers in solution and then monomerise before oligomerisation. This is a form of regulation. Since they do show binding to PA and PS-containing liposomes it can't be hard to also show that the protein then oligomerises, if it does. Can the authors propose a lipid-binding site on CelTOS?

3) Did the authors attempt microinjecting CelTOS in RAW or *Hep* G2 cells to demonstrate membrane rupture? In the context of malaria parasite infection, these cellular models are more relevant than the *Xenopus* oocyte system.

4) Other proteins are required for cell traversal in apicomplexan parasites, in particular perforin-like proteins (PLPs). Interestingly, expression and function of PLPs vary depending on the parasite stage. A recent paper (Risco-Castillo et al., 2015) has shown that Plasmodium sporozoites traverse cells within transient vacuoles, and the egress from these vacuoles requires PLP1. In contrast, how the parasite exits cells remained unknown, but the present study supports a crucial role of CelTOS. It thus appears that during cell traversal PLPs are required for breaching the membrane from the outer leaflet, whereas CelTOS would mediate membrane breaching from the inner leaflet. More generally, the parasites may use a range of PLPs proteins to disrupt the membrane of different host cell types, whereas a universal protein (CelTOS) would be used inside cells. Variations in membrane compositions (especially on the outer leaflet, where specific receptors might be involved in binding PLPs) may explain these specificities. The authors may consider including these considerations in the Discussion section and refine their model in Figure 7.

---

## [Author Response]

*Essential revisions:*

*1) The crystallographic work appears competently executed, yet insufficiently described for experiments to be repeated. For example, in the subsection “Crystallization, data collection, and structural studies”: Mercury and platinum derivative data, the authors should be more specific, and give more details (compound formula, concentration, soak duration).*

We thank the reviewers for indicating that the crystallographic work appears competently executed and for pointing out the need to include sufficient description to facilitate future repetition of experiments. The description of the procedure for obtaining crystals of the native, platinum- and mercury-derivatives of CelTOS have been included in the Methods section, and are described below:

“PvCelTOS crystals were grown at 17^o^C by hanging-drop vapor diffusion after mixing 1 µl of protein at 20 mg/ml with 1 µl of reservoir containing 1.6 M Ammonium dihydrogen phosphate, 0.08 M Tris pH 8.5 and 20% glycerol and streak seeding PvCelTOS crystals previously obtained from the same crystallization condition. […] Crystals of the CelTOS Mercury- derivative were obtained by soaking native CelTOS crystals for 1-2 minutes in cryo-protectant solutions containing: 1 µM mercury ethylphosphate, 1.6 M Ammonium dihydrogen phosphate, 0.08 M Tris pH 8.5 and 30% glycerol.”

*2) The authors interpretation of their work depends on CelTOS forming an oligomer on the membrane but they provide no data to show this. The protein is a dimer in the crystal and in solution (AUC data). Does it act as a dimer or do the authors think it comes apart to act? Some other pore-forming proteins are constitutive dimers in solution and then monomerise before oligomerisation. This is a form of regulation. Since they do show binding to PA and PS-containing liposomes it can't be hard to also show that the protein then oligomerises, if it does. Can the authors propose a lipid-binding site on CelTOS?*

We have provided additional experimental data to demonstrate that CelTOS forms pores of ~50 nm in diameter by transmission electron microscopy (Figure 5). The large diameter and shape of the pore strongly suggest the pore is stabilized by an oligomer of CelTOS. While we agree with the reviewers that changes in oligomeric state (dimer to monomer to oligomer transitions) are important regulatory mechanisms, deciphering these changes and the steps that precede pore formation as visualized by the new data will require several months to a year to accurately characterize. We therefore feel these studies are beyond the scope of the current manuscript. Never-the-less the additional electron microscopy data we now present strongly validates quantitative dextran blocking data and supports our model that CelTOS inserts in to membranes in a PA-specific manner to form large pores.

In addition, the reviewers request that we propose a lipid-binding site in CelTOS. While we strongly believe the basic patch on the surface of CelTOS may play a role in binding negatively charged phospholipids we feel this would be speculation at best (Figure 1). We therefore do not wish to over interpret the data at present and do not feel confident proposing an unambiguous lipid-binding site in CelTOS.

*3) Did the authors attempt microinjecting CelTOS in RAW or Hep G2 cells to demonstrate membrane rupture? In the context of malaria parasite infection, these cellular models are more relevant than the Xenopus oocyte system.*

We agree with the reviewer that RAW and HepG2 cells may be better cellular models for sporozoite traversal in the liver. *Xenopus oocytes* are an excellent model system for a generic cell to examine universal phenotypes of membrane disruption relevant to both the human host and mosquito vector. The requested experiment of microinjecting mammalian cells, which are far smaller and less robust than *Xenopus oocytes*, requires extensive technical expertise and specialized instrumentation neither of which is available nor can be performed in a timely manner. Further, we do not feel these additional experiments will provide any new insight not already obtained from the *Xenopus* microinjection experiments, and would only confirm the current experiments.

*4) Other proteins are required for cell traversal in apicomplexan parasites, in particular perforin-like proteins (PLPs). Interestingly, expression and function of PLPs vary depending on the parasite stage. A recent paper (Risco-Castillo et al., 2015) has shown that Plasmodium sporozoites traverse cells within transient vacuoles, and the egress from these vacuoles requires PLP1. In contrast, how the parasite exits cells remained unknown, but the present study supports a crucial role of CelTOS. It thus appears that during cell traversal PLPs are required for breaching the membrane from the outer leaflet, whereas CelTOS would mediate membrane breaching from the inner leaflet. More generally, the parasites may use a range of PLPs proteins to disrupt the membrane of different host cell types, whereas a universal protein (CelTOS) would be used inside cells. Variations in membrane compositions (especially on the outer leaflet, where specific receptors might be involved in binding PLPs) may explain these specificities. The authors may consider including these considerations in the Discussion section and refine their model in Figure 7.*

We thank the reviewers for providing constructive feedback to strengthen the manuscript. We have added the following to the manuscript to describe the relative functions of perforin-like proteins (PLPs) and CelTOS.

“The current literature supports a model where perforin-like proteins (PLPs) mediate entry into various pre-erythrocytic cells including the PLP-1 dependent disruption of transient vacuoles created during entry of hepatocytes (Risco-Castillo et al., 2015; Armstrong et al., 2004). This suggests a mechanism in which PLPs engage outer leaflet lipids for disruption although direct studies probing the biochemical function and lipid specificity of PLPs have not been described. Further, PLPs have been shown to play a role in egress of parasites from the parasitophorous vacuole and red blood cells (Tardieux and Ménard, 2008; Ishino, Chinzei and Yuda, 2005).*”*